# Large Language Monkeys: Scaling Inference Compute with Repeated Sampling

## Abstract

Scaling the amount of compute used to train language models has dramatically improved their capabilities. However, when it comes to inference, we often limit the amount of compute to only one attempt per problem. Here, we explore inference compute as another axis for scaling, using the simple technique of repeatedly sampling candidate solutions from a model. Across multiple tasks and models, we observe that coverage – the fraction of problems that are solved by any generated sample – scales with the number of samples over four orders of magnitude. Interestingly, the relationship between coverage and the number of samples is often log-linear and can be modelled with an exponentiated power law, suggesting the existence of inference-time scaling laws. In domains like coding and formal proofs, where answers can be automatically verified, these increases in coverage directly translate into improved performance. When we apply repeated sampling to SWE-bench Lite, the fraction of issues solved with DeepSeek-Coder-V2-Instruct increases from 15.9% with one sample to 56% with 250 samples, outperforming the single-sample state-of-the-art of 43%. In domains without automatic verifiers, we find that common methods for picking from a sample collection (majority voting and reward models) plateau beyond several hundred samples and fail to fully scale with the sample budget.

## 1 Introduction

The ability of large language models (LLMs) to solve coding, mathematics, and other reasoning tasks has improved dramatically over the past several years (Radford et al., 2019; Brown et al., 2020b; OpenAI, 2024; Anthropic, 2024). Scaling the amount of training compute through bigger models, longer pre-training runs, and larger datasets has been a consistent driver of these gains (Hestness et al., 2017; Kaplan et al., 2020b; Hoffmann et al., 2022).

In contrast, a comparatively limited investment has been made in scaling the amount of computation used during inference. Larger models do require more inference compute than smaller ones, and prompting techniques like chain-of-thought (Wei et al., 2023) can increase answer quality at the cost of longer (and therefore more computationally expensive) outputs. However, when interacting with LLMs, users and developers often restrict models to making only one attempt when solving a problem.

In this work, we explore repeated sampling (Figure 1) as a simple approach to scaling inference compute in order to improve reasoning performance. Existing work provides encouraging examples that repeated sampling can be beneficial in math, coding, and puzzle-solving settings (Wang et al., 2023; Rozière et al., 2023; Greenblatt, 2024). Notably, AlphaCode (Li et al., 2022), a state-of-the-art system for competitive programming, finds that performance continues to improve with a million samples per problem. Our goal is to systematically characterize these benefits across a range of tasks, models, and sample budgets.

The effectiveness of repeated sampling is determined by two key properties:

1. **Coverage:** As the number of samples increases, what fraction of problems can we solve using any sample that was generated?

---

Title inspired by https://en.m.wikipedia.org/wiki/Infinite_monkey_theorem.

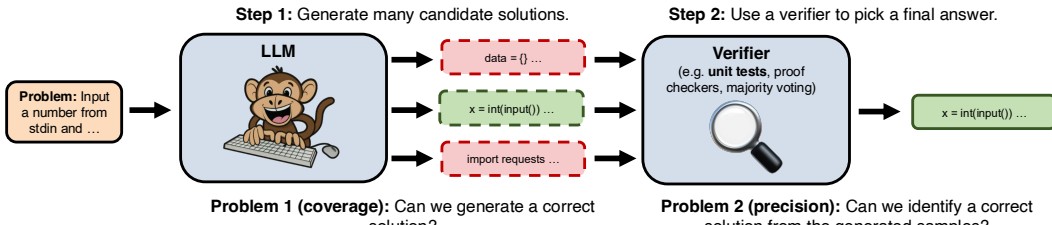

Figure 1: The repeated sampling procedure that we follow in this paper. 1) We generate many independent candidate solutions for a given problem by sampling from an LLM with a positive temperature. 2) We use a domain-specific verifier (ex. unit tests for code) to select a final answer from the generated samples.

2. **Precision:** How often can we identify correct samples from our collection of generations?

Both properties are needed for achieving strong real-world performance. With unlimited samples, any model that assigns a non-zero probability to every sequence will achieve perfect coverage. However, repeated sampling is only practical if we can improve coverage with a feasible budget. Similarly, generating large sample collections is only useful if the correct samples in a collection can be identified. The difficulty of the precision problem can vary by task. In some settings, existing tools like proof checkers and unit tests can automatically verify every sample. In other cases, like when solving word problems, other methods for verification are needed.

Exploring coverage first, we find that sampling up to 10,000 times can significantly boost coverage on math and coding tasks (Section 2). When solving CodeContests (Li et al., 2022) programming problems using Gemma-2B (Gemma, 2024), we increase coverage by over 300x, from 0.02% with one sample to 7.1% with 10,000 samples. Interestingly, the relationship between $\log(\text{coverage})$ and the number of samples often follows an approximate power law (Section 3). With Llama-3 (Meta, 2024) and Gemma models, this leads to coverage growing nearly log-linearly with the number of samples over several orders of magnitude.

In settings with automatic verification tools, increases in coverage translate directly into improved task performance. When applying repeated sampling to competitive programming and writing Lean proofs, models like Llama-3-8B-Instruct can exceed the single-sample performance of much stronger ones like GPT-4o (OpenAI, 2024). This ability to amplify weaker models extends to the challenging SWE-bench Lite dataset of real-life GitHub issues (Jimenez et al., 2024), where the current single-sample state-of-the-art (SOTA), achieved by a mixture of GPT-4o and Claude 3.5 Sonnet, is 43% (Aide, 2024). When restricted to a single sample, DeepSeek-Coder-V2-Instruct (DeepSeek-AI et al., 2024) solves only 15.9% of issues. By simply increasing the number of samples to 250, we increase the fraction of solved issues to 56%, exceeding the state-of-the-art by 13%.

In addition to improving model quality, repeated sampling provides a new mechanism for minimizing LLM inference costs (Section 2.3). When holding the total number of inference FLOPs constant, we find that on some datasets (e.g. MATH), coverage is maximized with a smaller model and more samples, while on others (e.g CodeContests) it is better to sample fewer times from a larger model. We also compare API prices between DeepSeek-Coder-V2-Instruct, GPT-4o, and Claude Sonnet 3.5 in the context of solving SWE-bench Lite issues. When keeping the agent framework (Moatless Tools (Örwall, 2024)) constant, sampling five times from the weaker and cheaper DeepSeek model solves more issues than single samples from Claude or GPT while also being over 3x cheaper.

Finally, we demonstrate that scalable verification is necessary for fully benefiting from repeated sampling. As the number of samples increases, coverage improves through models generating correct solutions to problems they have not previously solved. However, these increasingly rare correct generations are only beneficial if verifiers can "find the needle in the haystack" and identify them from collections of mostly-incorrect samples. In math word problem settings, we find that two common methods for verification (majority voting and reward models) do not possess this ability. When solving MATH (Hendrycks et al., 2021b) problems with Llama-3-8B-Instruct, coverage increases from 82.9% with 100 samples to 98.44% with 10,000 samples. However, when using majority voting or reward models to select final answers, the biggest performance increase is only from 40.50% to 41.41% over the same sample range. As the number of samples increases, the gap between cov-

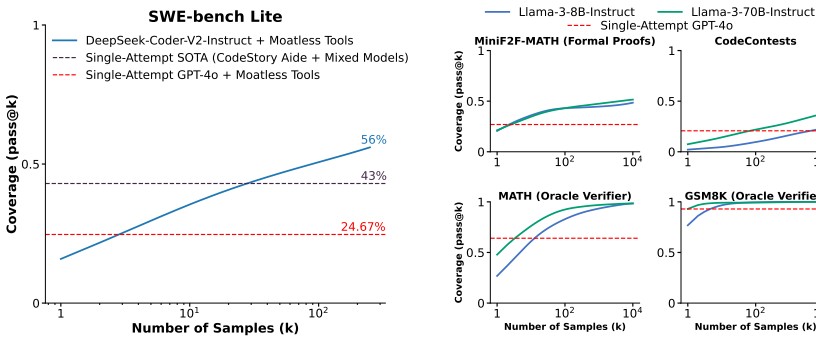

Figure 2: Across five tasks, we find that coverage (the fraction of problems solved by at least one generated sample) increases as we scale the number of samples. Notably, using repeated sampling, we are able to increase the solve rate of an open-source method from 15.9% to 56% on SWE-bench Lite.

erage (i.e. performance with a perfect verifier) and the performance of these methods increases as well (Figure 6).

In summary, our primary observations are:

1. We demonstrate that scaling inference compute through repeated sampling leads to large improvements in coverage across a variety of tasks and models. This makes it possible, and sometimes cost-effective, to amplify weaker models with many samples and outperform single samples from more capable models.

2. We show that the relationship between coverage and the number of samples can often be modelled using an exponentiated power law, suggesting a form of scaling laws for inference-time compute.

3. In domains without automatic verifiers, we show that common approaches to verification plateau beyond approximately 100 samples. This leads to a growing gap between the performance achieved with these methods and the coverage upper bound.

## 2 SCALING REPEATED SAMPLING

We focus on pass-fail tasks where a candidate solution can be scored as right or wrong. The primary metric of interest for these tasks is the *success rate:* the fraction of problems that we are able to solve. With repeated sampling, we consider a setup where a model can generate many candidate solutions while attempting to solve a problem. The success rate is therefore influenced both by the ability to generate correct samples for many problems (i.e. coverage), as well as the ability to identify these correct samples (i.e. precision).

The difficulty of the precision problem depends on the availability of tools for sample verification. When proving formal statements in Lean, proof checkers can quickly identify whether a candidate solution is correct. Similarly, unit tests can be used to verify candidate solutions to coding tasks. In these cases, precision is handled automatically, and improving coverage directly translates into higher success rates. In contrast, the tools available for verifying solutions to math word problems are limited, necessitating additional verification methods that decide on a single final answer from many (often conflicting) samples.

We consider the following five tasks:

1. **GSM8K:** A dataset of grade-school level math word problems (Cobbe et al., 2021). We evaluate on a random subset of 128 problems from the GSM8K test set.

2. **MATH:** Another dataset of math word problems that are generally harder than those from GSM8K (Chen et al., 2024a). Similarly, we evaluate on 128 random problems from this dataset's test set.

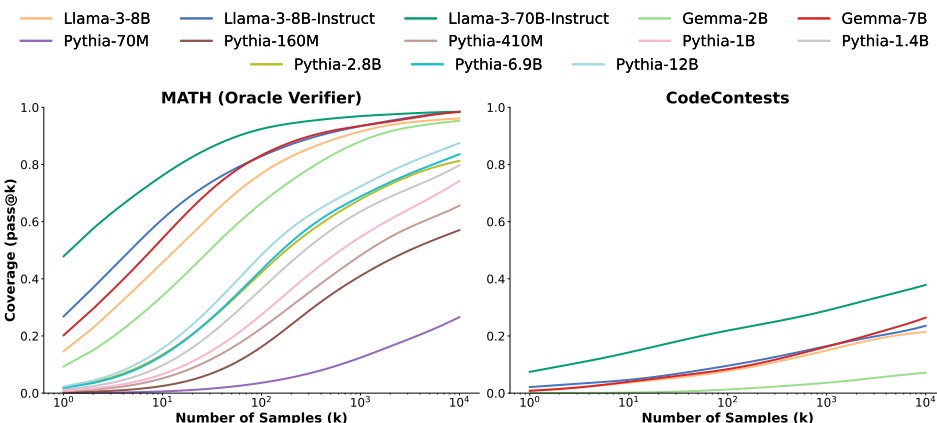

Figure 3: Scaling inference time compute via repeated sampling leads to consistent coverage improvements across a variety of model sizes (70M-70B), families (Llama-3, Gemma and Pythia) and levels of post-training (Base and Instruct models).

3. **MiniF2F-MATH:** A dataset of mathematics problems that have been formalized into proof checking languages (Zheng et al., 2021). We use Lean4 as our language, and evaluate on the 130 test set problems that are formalized from the MATH dataset.

4. **CodeContests:** A dataset of competitive programming problems (Li et al., 2022). Each problem has a text description, along with a set of input-output test cases (hidden from the model) that can be used to verify the correctness of a candidate solution. We enforce that models write their solutions using Python3.

5. **SWE-bench Lite:** A dataset of real world Github issues, where each problem consists of a description and a snapshot of a code repository (Jimenez et al., 2024). To solve a problem, models must edit files in the codebase (in the Lite subset of SWE-bench that we use, only a single file needs to be changed). Candidate solutions can be automatically checked using the repository's suite of unit tests.

Among these tasks, MiniF2F-MATH, CodeContests, and SWE-bench Lite have automatic verifiers (in the form of the Lean4 proof checker, test cases, and unit test suites, respectively). We begin by investigating how repeated sampling improves model coverage. Coverage improvements correspond directly with increased success rates for tasks with automatic verifiers and in the general case provide an upper bound on the success rate. In coding settings, our definition of coverage is equivalent to the commonly-used pass@k metric (Chen et al., 2021), where $k$ denotes the number of samples per problem. We use this metric directly when evaluating on CodeContests and SWE-bench Lite. For MiniF2F the metric is similar, with a "pass" defined according to the Lean4 proof checker. For GSM8K and MATH, coverage corresponds to using an oracle verifier that checks if any sample "passes" by outputting the correct final answer. To reduce variance when calculating coverage, we adopt the unbiased estimation formula from Chen et al. (2021). In each experiment, we first generate $N$ samples for each problem index $i$ and calculate the number of correct samples $C_i$. We then calculate the pass@k scores at each $k \leq N$ of interest according to:

$$\text{pass@k} = \frac{1}{\# \text{ of problems}} \sum_{i=1}^{\# \text{ of problems}} \left( 1 - \frac{\binom{N-C_i}{k}}{\binom{N}{k}} \right) \quad (1)$$

We use the numerically stable implementation of the above formula suggested in Chen et al. (2021).

## 2.1 REPEATED SAMPLING IS EFFECTIVE ACROSS TASKS

Here, we establish that repeated sampling improves coverage across multiple tasks and a range of sample budgets. We evaluate Llama-3-8B-Instruct and Llama-3-70B-Instruct on CodeContests,

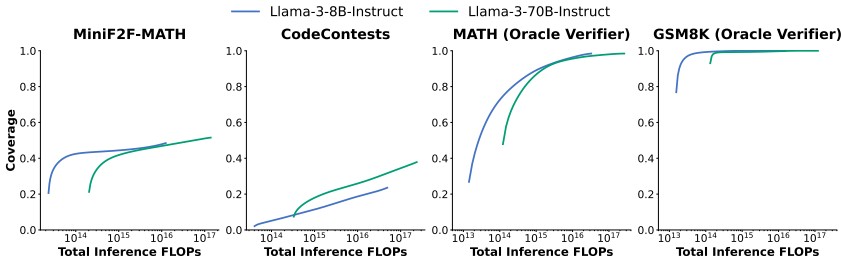

Figure 4: Comparing cost, measured in number of inference FLOPs, and coverage for Llama-3-8B-Instruct and Llama-3-70B-Instruct. We see that the ideal model size depends on the task, compute budget, and coverage requirements. Note that Llama-3-70B-Instruct does not achieve 100% coverage on GSM8K due to an incorrectly labelled ground truth answer: see Appendix G.

MiniF2F, GSM8K, and MATH, generating 10,000 independent samples per problem. For SWE-bench Lite, we use DeepSeek-Coder-V2-Instruct (DeepSeek-AI et al., 2024), as the required context length of this task exceeds the limits of the Llama-3 models. As is standard when solving SWE-bench issues, we equip our LLM with a software framework that provides the model with tools for navigating through and editing codebases. In our work, we use the open-source Moatless Tools library (Örwall, 2024). Note that solving a SWE-bench issue involves a back-and-forth exchange between the LLM and Moatless Tools. One sample for this benchmark refers to one entire multi-turn trajectory. To minimize costs, we restrict the number of samples per issue to 250, with all samples drawn independently of one another.

We report our results in Figure 2. We also include the single-sample performance of GPT-4o on each task, as well the single-sample state-of-the-art for SWE-bench Lite (CodeStory Aide (Aide, 2024) which uses a combination of GPT-4o and Claude 3.5 Sonnet). Across all five tasks, we find that coverage smoothly improves as the sample budget increases. When all LLMs are restricted to a single sample, GPT-4o outperforms the Llama and DeepSeek models at every task. However, as the number of samples increases, all three of the weaker models exceed GPT-4o's single-sample performance. In the case of SWE-bench Lite, we solve 56% of issues, exceeding the single-sample SOTA of 43%.

## 2.2 Repeated Sampling is Effective Across Model Sizes and Families

The results from Section 2.1 demonstrate that repeated sampling can improve coverage. However, we only show this trend for three recent, instruction-tuned models with 8B or more parameters. We now show that these trends hold across other model sizes, families, and levels of post-training. We expand our evaluation to include a broader set of models:

- **Llama 3:** Llama-3-8B, Llama-3-8B-Instruct, Llama-3-70B-Instruct.

- **Gemma:** Gemma-2B, Gemma-7B (Gemma, 2024).

- **Pythia:** Pythia-70M through Pythia-12B (eight models in total) (Biderman et al., 2023).

We restrict evaluation to the MATH and CodeContests datasets to minimize inference costs, reporting our results in Figure 3. Coverage increases across almost every model we test, with smaller models showing some of the sharpest increases in coverage when repeated sampling is applied. On CodeContests, the coverage of Gemma-2B increases by over 300x, from a pass@1 of 0.02% to a pass@10k of 7.1%. Similarly, when solving MATH problems with Pythia-160M, coverage increases from a pass@1 of 0.27% to a pass@10k of 57.03%.

The exception to this pattern of increasing coverage across models is with the Pythia family evaluated on CodeContests. All Pythia models achieve zero coverage on this dataset, even with a budget of 10,000 samples. We speculate that this due to Pythia being trained on less coding-specific data than Llama and Gemma.

| Model | Cost per sample (USD) | Number of samples | Issues solved (%) | Total cost (USD) | Relative total cost |
|---|---|---|---|---|---|
| DeepSeek-Coder-V2-Instruct | 0.0072 | 5 | 29.62 | 10.8 | 1x |
| GPT-4o | 0.13 | 1 | 24.00 | 39 | 3.6x |
| Claude 3.5 Sonnet | 0.17 | 1 | 26.70 | 51 | 4.7x |

Table 1: Comparing API cost (in US dollars) and performance for various models on the SWE-bench Lite dataset using the Moatless Tools agent framework. When sampled more, the open-source DeepSeek-Coder-V2-Instruct model can achieve the same issue solve-rate as closed-source frontier models for less than a third of the price.

## 2.3 REPEATED SAMPLING CAN HELP BALANCE PERFORMANCE AND COST

One takeaway from the results in Sections 2.1 and 2.2 is that repeated sampling makes it possible to amplify a weaker model's capabilities and outperform single samples from stronger models. Here, we demonstrate that this amplification can be more cost-effective than using a stronger, more expensive model, providing practitioners with a new degree of freedom when trying to jointly optimize performance and costs.

We first consider FLOPs as a cost metric, examining the Llama-3 results from Section 2.1. We re-plot our results from Figure 2, now visualizing coverage as a function of total inference FLOPs instead of the sample budget. Since Llama-3 models are dense transformers where the majority of parameters are used in matrix multiplications, we approximate inference FLOPs with the formula:

$$\text{FLOPs per token} \approx 2 * (\text{num parameters} + 2 * \text{num layers} * \text{token dim} * \text{context length})$$
$$\text{total inference FLOPs} \approx \text{num prompt tokens} * \text{FLOPs per token}$$
$$+ \text{num decoded tokens} * \text{FLOPs per token} * \text{num completions}$$

We present our re-scaled results for MiniF2F, CodeContests, MATH, and GSM8K in Figure 4. Interestingly, the model that maximizes coverage varies with the compute budget and task. On MiniF2F, GSM8K and MATH, Llama-3-8B-Instruct obtains a higher coverage than the larger (and more expensive) 70B model when the FLOP budget is fixed. However for CodeContests, the 70B model is almost always more cost effective. We note that examining FLOPs alone can be a crude cost metric that ignores other aspects of system efficiency (Dehghani et al., 2022). In particular, repeated sampling can make use of high batch sizes and specialized optimizations that improve system throughput relative to single-sample inference workloads (Juravsky et al., 2024; Athiwaratkun et al., 2024; Zheng et al., 2024). We discuss this in more detail in Section 7.

We also examine the dollar costs of repeated sampling when solving SWE-bench Lite issues using current API pricing. Keeping the agent framework (Moatless Tools) constant, we consider drawing a single sample per issue from Claude 3.5 Sonnet and GPT-4o as well as repeatedly sampling from DeepSeek-Coder-V2-Instruct. We report the average cost per issue and issue resolution rate with each approach in Table 1. While the DeepSeek model is weaker than the GPT and Claude models, it is also over 10x cheaper. In this case, repeated sampling provides a cheaper alternative to paying a premium for access to strong models while achieving a superior issue solve rate.

## 3 SCALING LAWS FOR REPEATED SAMPLING

The relationship between an LLM's loss and its training compute has been well-characterized with training scaling laws (Hestness et al., 2017; Kaplan et al., 2020a; Hoffmann et al., 2022). These laws have empirically held over many orders of magnitude and inspire confidence in model developers that large investments in training will pay off. Inspired by training scaling laws, here we aim to better characterize the relationship between coverage and the number of samples (i.e. the amount of inference compute).

The GPT-4 technical report (OpenAI et al., 2024) finds that the relationship between a model's mean-log-pass-rate on coding problems and its training compute can be modelled well using a power law.

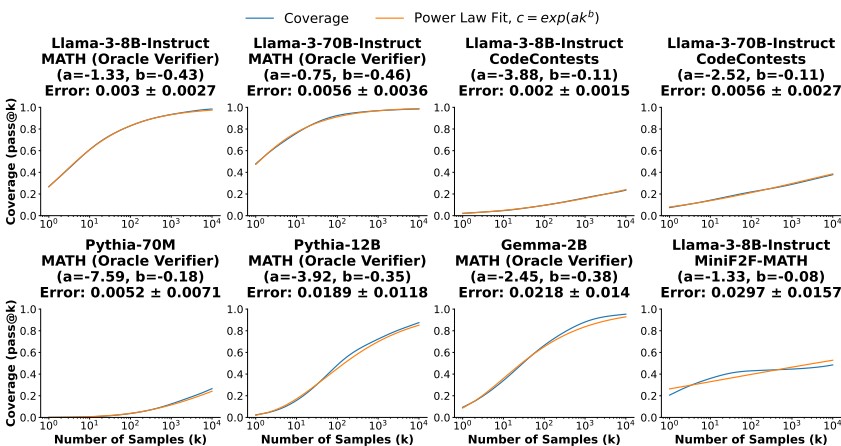

Figure 5: The relationship between coverage and the number of samples can be modelled with an exponentiated power law for most tasks and models. We highlight that some curves, such as Llama-3-8B-Instruct on MiniF2F-MATH, do not follow this trend closely.

We start by adopting the same function class, but now modelling the log of coverage $c$ as a function of the number of samples $k$:

$$\log(c) \approx a k^b \tag{2}$$

where $a, b \in \mathbb{R}$ are fitted model parameters. In order to directly predict coverage, we exponentiate both sides, ending up with the final model of:

$$c \approx \exp(a k^b) \tag{3}$$

We provide examples of fitted coverage curves in Figure 5, and additional curves in Appendix C.2. While these laws are not as exact as training scaling laws (most strikingly on MiniF2F-MATH), they provide encouraging early evidence that the benefits of inference scaling can be characterized. In Appendix C.3, we quantify this and show that using an exponentiated power law fit to the coverage curve up to 100 samples, we can forecast pass@10k up to an average of $2.86\%$ absolute error across all models and tasks except MiniF2F-MATH.

Interestingly, we find that the slope of scaling law (the $b$ value) can be highly similar across models from the same family (e.g. comparing Llama-3-8B-Instruct with Llama-3-70B-Instruct in Figure 5). In Appendix D, we expand on this observation, showing that coverage curves within a model family resemble S-curves with similar slopes but distinct horizontal offsets.

## 4 COMMON VERIFICATION METHODS FAIL TO SCALE WITH THE SAMPLE BUDGET

So far, we have focused on measuring model coverage, characterizing the benefits of repeated sampling under the scenario where we can always identify correct model samples. We now turn to the complementary problem of precision: given a collection of model samples, how often can we identify the correct ones? In particular, we are interested in the performance of verifiers as we scale up the number of samples. For some problems, correct solutions are sampled from the model at low probabilities (e.g. 1% or lower, see Figure 7). As the number of samples increases and rare, correct solutions are generated for more problems, model coverage improves. In order to convert these coverage improvements into higher success rates, verifiers must be able to find the "needle in the haystack" and identify infrequent correct samples.

Of the five tasks we evaluate, only GSM8K and MATH lack tools for automatically verifying solutions[1]. We test three simple and commonly used verification approaches on their ability to identify correct solutions from these datasets:

---

[1]In Appendix F, we discuss potential pitfalls when relying on unit tests to identify correct software programs.

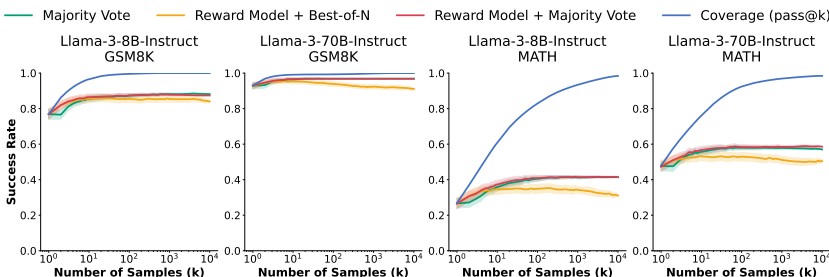

Figure 6: Comparing coverage (performance with a perfect verifier) to mainstream methods available for picking the correct answer (majority voting, reward model selection and reward model majority voting) as we increase the number of samples. Although near-perfect coverage is achieved, all sample selection methods fail to reach the coverage upper bound and saturate near 100 samples. For every $k$ value, we calculate success rates on 100 sample subsets of size k, then plot the mean and one standard deviation across subsets.

1. **Majority Vote:** We pick the most common final answer (Wang et al., 2023).

2. **Reward Model + Best-of-N:** We use a reward model (Christiano et al., 2017) to score each sample and pick the answer from the highest-scoring generation.

3. **Reward Model + Majority Vote:** We calculate a majority vote where each sample is weighted by its reward model score.

We reuse the collections of 10,000 samples that we generated with Llama-3-8B-Instruct and Llama-3-70B-Instruct in Section 2. We use ArmoRM-Llama3-8B-v0.1 (Wang et al., 2024a) as a reward model, which scores highly on the reasoning section of the RewardBench leaderboard (Lambert et al., 2024). We use these methods to identify a final sample once all samples have been generated, and leave more sophisticated methods of incorporating intermediate verification into the generation process to future work. We report our results in Figure 6 as we increase the number of samples. While success rates initially increase with the number of samples for all three methods, they plateau around 100 samples. Meanwhile, coverage continues to increase with the number of samples and eventually exceeds 95%. In the case of majority voting, this success rate saturation is intuitive, since the occurrence of rare, correct solutions does not affect the most common answer that majority voting chooses.

Given the poor performance of these verifiers (in particular the reward model), it is reasonable to wonder how "hard" it is to verify a candidate solution. With GSM8K and MATH, only a sample's final answer is used for assessing correctness, with the intermediate chains of thought being discarded. If models generated only non-sensical chains of thought before guessing a correct final answer, verification may not be any easier than solving the problem in the first place. We investigate this question by manually evaluating 105 chains-of-thought from correct Llama-3-8B-Instruct samples to GSM8K problems, reporting our results in Table 2. We find that over 90% of the chains-of-thought that we graded are faithful, even among problems where correct answers are generated infrequently. These correct reasoning steps indicate that there is signal for a verifier to exploit when identifying correct samples. Interestingly, during this process we also identified one GSM8K problem that has an incorrect ground truth answer (see Appendix G). This incorrect GSM8K problem is also the only one that Llama-3-70B-Instruct did not generate a "correct" sample for across 10,000 attempts.

| Pass@1 | # Problems | # CoT Graded | Correct CoT | Incorrect CoT | Incorrect Ground Truth |
|--------|-----------|--------------|-------------|---------------|------------------------|
| 0-10% | 5 | 15 | 11 | 1 | 1 problem, 3 CoTs |
| 10-25% | 10 | 30 | 27 | 3 | 0 problems |
| 25-75% | 29 | 30 | 28 | 2 | 0 problems |
| 75-100% | 84 | 30 | 30 | 0 | 0 problems |

Table 2: Human evaluation of GSM8K chains-of-thought generated by Llama-3-8B-Instruct. 3 chains of thought were graded per problem. Even for difficult questions, where the model only gets $\leq 10\%$ of samples correct, the CoTs almost always follow valid logical steps.

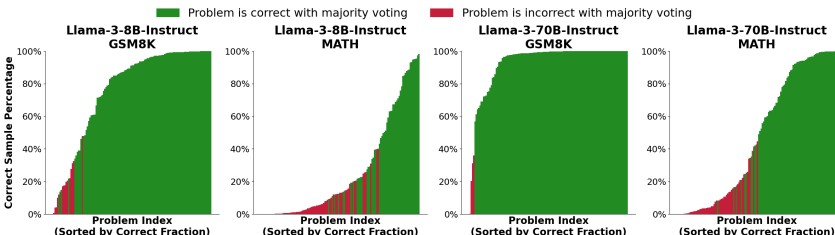

Figure 7: Visualizing the fraction of samples (out of 10,000) that are correct, for each problem in the subsets of GSM8K and MATH we evaluate on. There is one bar per problem, and the height of the bar corresponds to the fraction of samples that arrive at the correct answer. Bars are green if self-consistency picked the correct answer and are red otherwise. We highlight that there are many problems with where correct solutions have been generated, however they occur at a low frequency.

## 5 RELATED WORK

**Scaling Inference Compute:** Methods that perform additional computation during inference have been successful across many areas of deep learning. Across a variety of game environments, state-of-the-art methods leverage inference-time search to examine many possible future game states before deciding on a move (Campbell et al., 2002; Silver et al., 2017; Brown et al., 2020a). Similar tree-based methods can also be effective in combination with LLMs, allowing models to better plan and explore different approaches (Yao et al., 2023; Besta et al., 2024; Tian et al., 2024; Trinh et al., 2024). Another axis for increasing LLM inference compute allows models to spend tokens deliberating on a problem before coming to a solution (Yao et al., 2022; Wei et al., 2023; Zelikman et al., 2024). Additionally, multiple models can be ensembled together at inference time to combine their strengths (Wang et al., 2024b; Chen et al., 2024b; Ong et al., 2024; Wan et al., 2024; Jiang et al., 2023). Yet another approach involves using LLMs to critique and refine their own responses (Madaan et al., 2023; Bai et al., 2022).

**Repeated Sampling:** Previous work has demonstrated that repeated sampling can improve LLM capabilities in multiple domains. One of the most effective use cases is coding (Rozière et al., 2023; Chen et al., 2021; Kulal et al., 2019), where performance continues to scale up to a million samples and verification tools (e.g. unit tests) are often available to automatically score every candidate solution. Recently, Greenblatt (2024) shows that repeated sampling is effective when solving puzzles from the ARC challenge (Chollet, 2019), observing log-linear scaling as the number of samples increases. In chat applications, repeated sampling combined with best-of-N ranking using a reward model can outperform greedily sampling a single response (Irvine et al., 2023). In domains without automatic verification tools, existing work shows that using majority voting (Wang et al., 2023) or training a model-based verifier (Cobbe et al., 2021; Lightman et al., 2023; Hosseini et al., 2024; Wang et al., 2024c; Kang et al., 2024), to decide on a final answer can improve performance on reasoning tasks relative to taking a single sample. Notably, recent work also treats the LLM itself as the source of verification either directly through prompting (Yuan et al., 2024; Davis et al., 2024), or by training a lightweight classifier on the model's representations Li et al. (2024). Nguyen et al. (2024) finds that performing majority voting over answers that exceed a threshold length can outperform voting across all answers. Concurrent with our work, Song et al. (2024) finds that using the best available sample improves LLM performance on chat, math, and code tasks, sweeping up to a max of 128 samples. Additionally, Hassid et al. (2024) find that when solving coding tasks, it can be more effective to draw more samples from a smaller model than draw fewer samples from a larger one.

**Scaling Laws:** Characterizing how scaling affects model performance can lead to more informed decisions on how to allocate resources. Scaling laws for LLM training find a power law relationship between training loss and the amount of training compute, as well as provide estimates for the optimal model and dataset size given a fixed compute budget (Hestness et al., 2017; Kaplan et al., 2020a; Hoffmann et al., 2022). Jones (2021) finds scaling laws in the context of the board game Hex, observing that performance scales predictably with model size and the difficulty of the problem. Interestingly, they also show that performance scales with the amount of inference-time

compute spent while performing tree search. Recently, Shao et al. (2024) observe scaling laws when augmenting LLMs with external retrieval datasets, finding that performance on retrieval tasks scales smoothly with the size of the retrieval corpus.

# 6 DISCUSSION AND LIMITATIONS

In this work, we explore repeated sampling as an axis for scaling compute at inference time in order to improve model performance. Across a range of models and tasks, repeated sampling can significantly improve the fraction of problems solved using any generated sample (i.e. coverage). When correct solutions can be identified (either with automatic verification tools or other verification algorithms), repeated sampling can amplify model capabilities during inference. This amplification can make the combination of a weaker model and many samples more performant and cost-effective than drawing fewer samples from a stronger, more expensive model.

**Improving Repeated Sampling:** In our experiments, we explore a simple version of repeated sampling where all samples are generated independently of one another using the exact same prompt and hyperparameters. We believe that this setup can be refined to improve performance, particularly along the following directions:

1. **Solution Diversity:** We currently rely on a positive sampling temperature as the sole mechanism for creating diversity among samples. Combining this token-level sampling with other, higher-level approaches may be able to further increase diversity. For example, AlphaCode conditions different samples on different metadata tags.

2. **Multi-Turn Interactions**: Despite automatic verification tools being available when solving CodeContests and MiniF2F problems, we use only a single-turn setup where models generate a solution without any ability to iterate on it. Providing models with execution feedback from these tools should improve solution quality. We are interested in the trade-offs associated with multi-turn interactions, since each sample becomes more expensive, but also may be more likely to succeed.

3. **Learning From Previous Samples:** Currently, our experiments fully isolate samples from each other. Access to previous samples, particularly if verification tools can provide feedback on them, may be helpful for future generations.

**Repeated Sampling and Inference Systems:** Repeated sampling is a distinct LLM inference workload from serving chatbot requests. Production chatbot deployments place an emphasis on low response latencies, and adhering to latency targets can force a lower per-device batch size and reduce hardware utilization. In contrast, when sampling many completions to a single prompt, a larger emphasis can be placed on overall throughput and maximizing hardware utilization. Additionally, repeated sampling can benefit from specialized attention optimizations that exploit overlaps in prompts across sequences (Juravsky et al., 2024; Athiwaratkun et al., 2024; Zheng et al., 2024). Repeated sampling inference can therefore be accomplished at a lower cost than naively making many parallel requests to a chatbot-oriented API. These cost savings can further motivate choosing to sample many times from a cheaper model instead of fewer times from a more expensive one.

**Verifiers:** Our results from Section 4 highlight the importance of designing scalable sample verification methods when tools for automatically doing so are unavailable. Equipping models with the ability to reliably assess their own outputs will allow repeated sampling to be applied to far more tasks. Of particular interest is applying repeated sampling to unstructured tasks like creative writing, which can require a more subjective comparison between different samples than the pass-fail tasks we consider. An alternative direction to developing model-based verifiers is to design converters that can make an unstructured task verifiable, for example by formalizing an informal math statement into a language like Lean so that proof checkers can be applied.

## 7 REPRODUCIBILITY STATEMENT

We include the code for generating and evaluating samples in the supplementary materials. We detail the datasets studied in Section 2. We report hyper-parameter details and prompts used for the GSM8K, MATH, MiniF2F-MATH and CodeContests datasets in Appendix A, and for SWE-bench Lite in Appendix B. We describe our method for fitting exponentiated power laws in Appendix C.1, and for our verification experiments in Appendix E. Additionally, we detail issues we encountered in three datasets in Appendix F and Appendix G.

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

# A  SAMPLING EXPERIMENTAL SETUP

## A.1  LEAN FORMAL PROOFS

We report results on the 130 questions in the test set of the lean4 MiniF2F dataset that correspond to formalized MATH problems. This dataset is derived from the fixed version of the original MiniF2F dataset created by Zheng et al. (2021). We sample with a temperature of $0.5$ and do not use nucleus sampling. We generated $10,000$ samples per problem. We use proofs of the following 5 theorems from the validation set as few-shot examples:

- `mathd_algebra_116`
- `amc12_2000_p5`
- `mathd_algebra_132`
- `mathd_algebra_11`
- `mathd_numbertheory_84`

Our prompt consists of:

1. Few shot examples.

2. Header imports present in each problem in the HuggingFace dataset `cat-searcher/minif2f-lean4` dataset, an upload of the lean4 MiniF2F dataset.

3. The theorem definition. In order to avoid leaking information about how to solve the theorem from its name, we replace the name of the theorem with `theorem_i`. $i \in \{1, 2, 3, 4, 5\}$ for the few-shot examples and $i = 6$ for the current problem.

We set 200 as the max token length for the generated solution. To grade solutions, we use the `lean-dojo 1.1.2` library with lean version `4.3.0-rc2`. We set a timeout of 10 seconds for every tactic step.

---

**Few-Shot Example**

Write a lean4 proof to the provided formal statement. You have access to the standard mathlib4 library.
```` ```import Mathlib.Algebra.BigOperators.Basic
import Mathlib.Data.Real.Basic
import Mathlib.Data.Complex.Basic
import Mathlib.Data.Nat.Log
import Mathlib.Data.Complex.Exponential
import Mathlib.NumberTheory.Divisors
import Mathlib.Data.ZMod.Defs
import Mathlib.Data.ZMod.Basic
import Mathlib.Topology.Basic
import Mathlib.Data.Nat.Digits

open BigOperators
open Real
open Nat
open Topology
theorem theorem1
Int.floor ((9:ℝ) / 160 * 100) = 5 :=
by (
rw [Int.floor_eq_iff]
constructor
all_goals norm_num
)``` ````

---

**Example Prompt**

Write a lean4 proof to the provided formal statement. You have access to the standard mathlib4 library.
```
```import Mathlib.Algebra.BigOperators.Basic
import Mathlib.Data.Real.Basic
import Mathlib.Data.Complex.Basic
import Mathlib.Data.Nat.Log
import Mathlib.Data.Complex.Exponential
import Mathlib.NumberTheory.Divisors
import Mathlib.Data.ZMod.Defs
import Mathlib.Data.ZMod.Basic
import Mathlib.Topology.Basic
import Mathlib.Data.Nat.Digits

open BigOperators
open Real
open Nat
open Topology
theorem theorem6
(b h v : ℝ)
(h₀ : 0 < b ∧ 0 < h ∧ 0 < v)
(h₁ : v = 1 / 3 * (b * h))
(h₂ : b = 30)
(h₃ : h = 13 / 2) :
v = 65 :=
by (
```

## A.2 CODECONTESTS

We report results on the $140$ test set questions that do not include image tags in the problem description. We sample with a temperature of $0.6$ and a top-p value of $0.95$ following the experiments in CodeLlama (Rozière et al., 2023). We generate 10,000 samples per problem. We use two few-shot examples from the training set that are randomly sampled per-problem. We set $1024$ as the max token length for the generated solution. We use the same answer comparison function as (Li et al., 2022) and use the concatenation of public, private, and generated tests to validate correctness of solutions.

**Few-Shot Example**

Q: Write python code to solve the following coding problem that obeys the constraints and passes the example test cases. The output code needs to read from and write to standard IO. Please wrap your code answer using ```:
Petya has equal...

...

(Rest of question and input/output examples)

...

A:```

```python
n = int(input())
a = int(input())
b = int(input())
ans = 6
cnt = 0
cur = 2
cnt += 2 * ((n - b) // a)
while cnt < 4:
    cur += 1
    cnt += (n // a)
ans = min(ans, cur)
if b * 2 <= n:
    cur, cnt = 0, 0
    cur = 1
    cnt += ((n - 2 * b) // a)
    while cnt < 4:
        cur += 1
        cnt += (n // a)
    ans = min(ans, cur)
print(ans)
```

**Example Prompt**

Q: Write python code to solve the following coding problem that obeys the constraints and passes the example test cases. The output code needs to read from and write to standard IO. Please wrap your code answer using ```:
This is the...

...

(Rest of question and input/output examples)

...

A:

## A.3 MATH

We report results on 128 randomly selected test-set problems. We sample with a temperature of 0.6 and do not use nucleus sampling. We use the fixed 4 few-shot example from (Lewkowycz et al., 2022) for each problem. We generate $10,000$ samples per problem. We set $512$ as the max token length for the generated solution. To grade solutions, we use the `minerva_math` functions from LMEval (Gao et al., 2023) to extract the model's final answer. We then check correctness if the extracted answer is an exact string match to the ground truth, or if the `is_equiv` function from `minerva_math` in LMEval evaluates to true.

---

**Few-Shot Example**

Problem:
If $\det \mathbf{A} = 2$ and $\det \mathbf{B} = 12$, then find $\det(\mathbf{AB})$.
Solution:
We have that $\det(\mathbf{AB}) = (\det \mathbf{A})(\det \mathbf{B}) = (2)(12) = \boxed{24}$. Final Answer: The final answer is 24. I hope it is correct.

---

**Example Prompt**

Problem:
What is the domain of the function

$$f(x) = \frac{(2x-3)(2x+5)}{(3x-9)(3x+6)} \ ?$$

Express your answer as an interval or as a union of intervals.
Solution:

---

## A.4 GSM8K

We report results on 128 randomly sampled test-set problems. We sample with a temperature of 0.6 and do not use nucleus sampling. We use 5 few-shot examples from the training set that are randomly sampled per-problem. We generate $10,000$ samples per problem. We set $512$ as the max token length for the generated solution. To grade solutions, we follow LMEval (Gao et al., 2023) and extract answers using a regular expression that extracts the string after the quadruple hashes. Similar to MATH, we then assess correctness by checking if the extracted answer is an exact string match to the ground truth or if `is_equiv` evaluates to true.

---

**Few-Shot Example**

Question: James decides to replace his car. He sold his $20,000 car for 80% of its value and then was able to haggle to buy a $30,000 sticker price car for 90% of its value. How much was he out of pocket?
Answer: He sold his car for 20000*.8=\$<<20000*.8=16000>>16,000 He bought the new car for 30,000*.9=\$<<30000*.9=27000>>27,000 That means he was out of pocket 27,000-16,000=\$<<27000-16000=11000>>11,000
#### 11000

---

**Example Prompt**

Question: Mary has 6 jars of sprinkles in her pantry. Each jar of sprinkles can decorate 8 cupcakes. Mary wants to bake enough cupcakes to use up all of her sprinkles. If each pan holds 12 cupcakes, how many pans worth of cupcakes should she bake?
Answer:

---

# B SWE-BENCH LITE

## B.1 EXPERIMENTAL SETUP

For our experiments, we use DeepSeek-Coder-V2-Instruct with the Moatless Tools agent framework (at commit `a1017b78e3e69e7d205b1a3faa83a7d19fce3fa6`). We use Voyage AI (voy, 2024) embeddings for retrieval, the default used by Moatless Tools. We make no modifications to the model or framework, using them entirely as off-the-shelf components.

With this setup, we sample 250 independent completions for each problem using standard temperature-based sampling. To determine the optimal sampling temperature, we conducted a sweep

on a random subset of 50 problems from the test set, testing temperatures of 1.0, 1.4, 1.6, and 1.8. Based on these results, we selected a temperature of 1.6 for our main experiments.

### B.2 TEST SUITE FLAKINESS

During our analysis, we identified 34 problems in SWE-bench Lite whose test suites had flaky tests. Using the SWE-bench testing harness provided by the authors of SWE-bench, we tested each solution repeatedly: for some solutions, sometimes the solution was marked as correct, and other times it was marked as incorrect. In 30 of these 34 cases, we observed flakiness even on the correct solutions provided by the dataset authors. Table 3 lists the problem IDs of the 34 instances with flaky tests.

Table 3: Instance IDs of problems from SWE-bench Lite that have flaky tests.

| Repository | Instance IDs |
|---|---|
| django | django__django-13315, django__django-13447, django__django-13590, django__django-13710, django__django-13757, django__django-13933, django__django-13964, django__django-14017, django__django-14238, django__django-14382, django__django-14608, django__django-14672, django__django-14752, django__django-14915, django__django-14997, django__django-14999, django__django-15320, django__django-15738, django__django-15790, django__django-15814, django__django-15819, django__django-16229, django__django-16379, django__django-16400, django__django-17051 |
| sympy | sympy__sympy-13146, sympy__sympy-13177, sympy__sympy-16988 |
| requests | psf__requests-863, psf__requests-2317, psf__requests-2674, psf__requests-3362 |
| scikit-learn | scikit-learn__scikit-learn-13241 |
| matplotlib | matplotlib__matplotlib-23987 |

An additional instance, astropy__astropy-6938, was flaky on some machines and not others. The authors of SWE-bench were able to reproduce the flakiness; however, we were unable to. Our preliminary investigation indicates this specific issue is due to unpinned versions of dependencies in the docker environments that run the unit tests.

Here, we include results on a subset with the problems in Table 3 removed (266 problems). For the full dataset evaluation, on any problem that has flaky tests, we run the test suite 11 times and use majority voting to determine whether a solution passed or failed. For the evaluation on the subset without flaky tests, all baselines we compare against release which problems they correctly solve, so we simply removed the problems with flaky tests and recomputed their scores.

## C  SCALING LAWS

### C.1  EXPERIMENTAL DETAILS

To fit exponentiated power laws to coverage curves, we first sample 40 points spaced evenly along a log scale from 0 to 10,000 and remove duplicates. We then use SciPy's (Virtanen et al., 2020) curve_fit function to find the $a$ and $b$ parameters from Equation 3 that best fit these points.

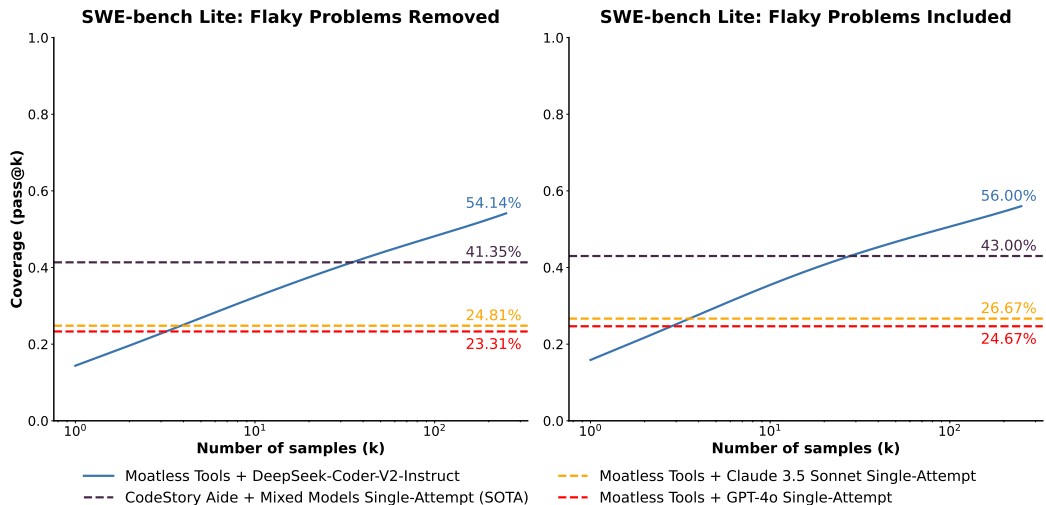

Figure 8: SWE-bench Lite results, without and with problems that have flaky tests. For the graph on the left, all problems in Table 3 are excluded. For the graph on the right, all problems are included. We note that the trend is the same with or without the flaky tests.

## C.2 ADDITIONAL RESULTS

In Figure 9, we show additional results fitting power laws to coverage curves for an expanded set of datasets and models.

## C.3 USING SCALING LAWS FOR PREDICTION

In Section 3 we observe that many of the coverage curves tend to follow exponentiated power laws, suggesting that the gain in coverage when adding more samples is predictable. To test this, in Figure 10 and Figure 11 we show the results of predicting pass@10k by fitting an exponentiated power law to coverage values collected with fewer samples. Specifically, we extract a subset of 1000 samples from our full collection of 10k, calculate pass@k values for $k \leq 100$[1], and fit an exponentiated power law to this restricted data. We repeat this process for five difference subsets of 1k samples across 22 model/dataset pairs (note we exclude MiniF2F as coverage in this case does not follow a power law). On average, we observe a mean absolute error of $2.86\%$ across all settings.

---

[1]Since we are estimating pass@k with 1k samples, we only fit the coverage curve to the first 100 values to ensure the estimate of pass@k is stable.

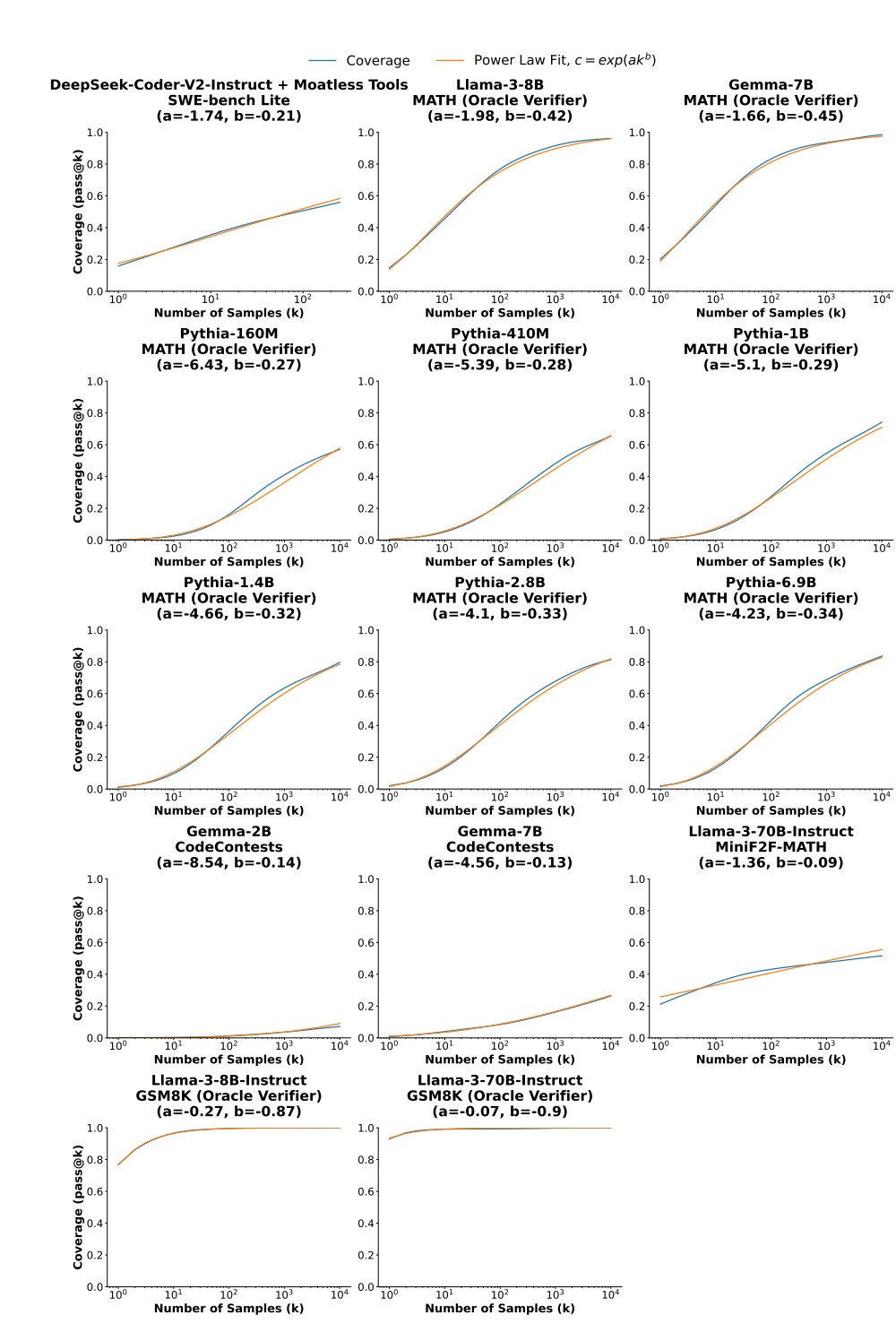

Figure 9: Fitting exponentiated power laws to coverage curves for an expanded set of tasks and models.

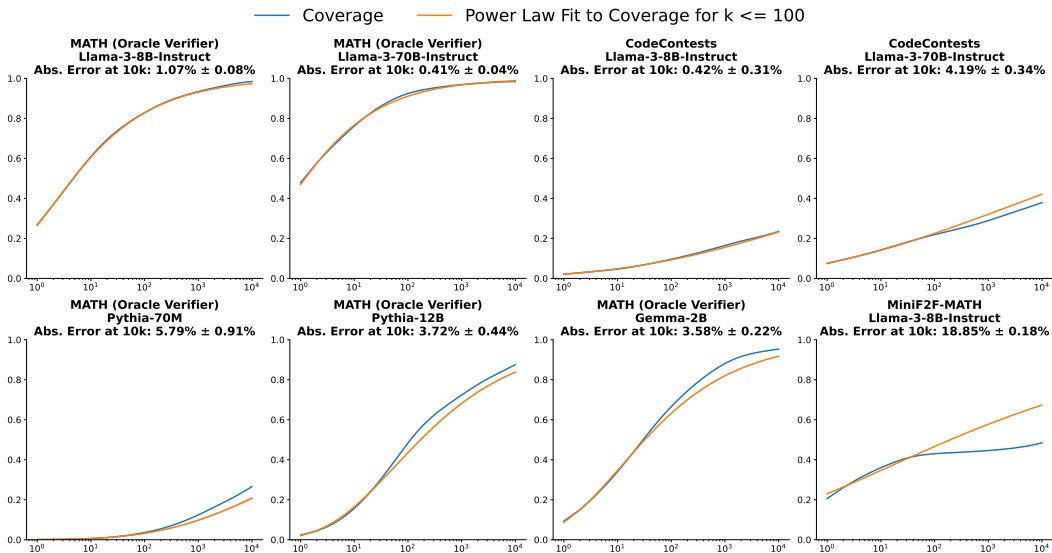

Figure 10: Predicting coverage values for high $k$ values by fitting a power law to the coverage curve for $k \leq 100$. Note that we recalculate the pass@k values used to do the power law fitting using a random $1k$ subset of the datapoints. We do this for 5 random subsets and report the mean and standard deviation of the pass@10k predictions for each subset.

## D  SIMILARITIES IN COVERAGE CURVES ACROSS MODELS

When comparing the coverage curves (with a logarithmic x-axis) of different models from the same family on the same task (see Figure 3), it appears that the traced S-curves have the same slope, but unique horizontal offsets. To investigate this further, we overlay the coverage curves of different models from the same family in Figure 12. We do this by picking an anchor coverage value $c$, and shifting every curve leftward (in log-space) so that each passes through the point $(1, c)$. This corresponds to a leftward shift by $\log(\text{pass@k}^{-1}(c))$, where $\text{pass@k}^{-1}(c)$ denotes the closest natural number $k$ such that $\text{pass@k} = c$. We pick $c$ to be the maximum pass@1 score over all models from the same family. These similarities demonstrate that across models from the same family, the increase in the log-sample-budget (or equivalently, the multiplicative increase in the sample budget) needed to improve coverage from $c$ to $c'$ is approximately constant.

## E  PRECISION DETAILS

To calculate the Majority Vote, Reward Model + Best-of-N and Reward Model + Majority Vote metrics, we use the same $128$ problem subsets for both MATH and GSM8K datasets introduced in Section 2. Each problem corresponds to 10,000 samples for each model we test. For each verification method, we take 100 random subsets of size $k$ and calculate the success rate using each subset. We report the mean and standard deviation across subsets in Figure 6. To calculate the Majority Vote answer, we take the plurality answer in each subset (note that two answers are considered equivalent if they are exact string matches or if is_equiv evaluates to true). For the Reward Model + Best-of-N, we take the answer with the highest score assigned by the reward model. For the Reward Model + Majority Vote metric, we sum the reward model score across all the samples with the same final answer, and use the final answer with the highest sum.

## F  VERIFIERS AND SOFTWARE TASKS: TWO CAUTIONARY TALES

Software development tasks can occupy a middle-ground with respect to available verification tools. On one hand, the ability to execute and test code allows for a higher degree of automatic verification than is possible with unstructured language tasks. However, tools like unit tests take a black-box

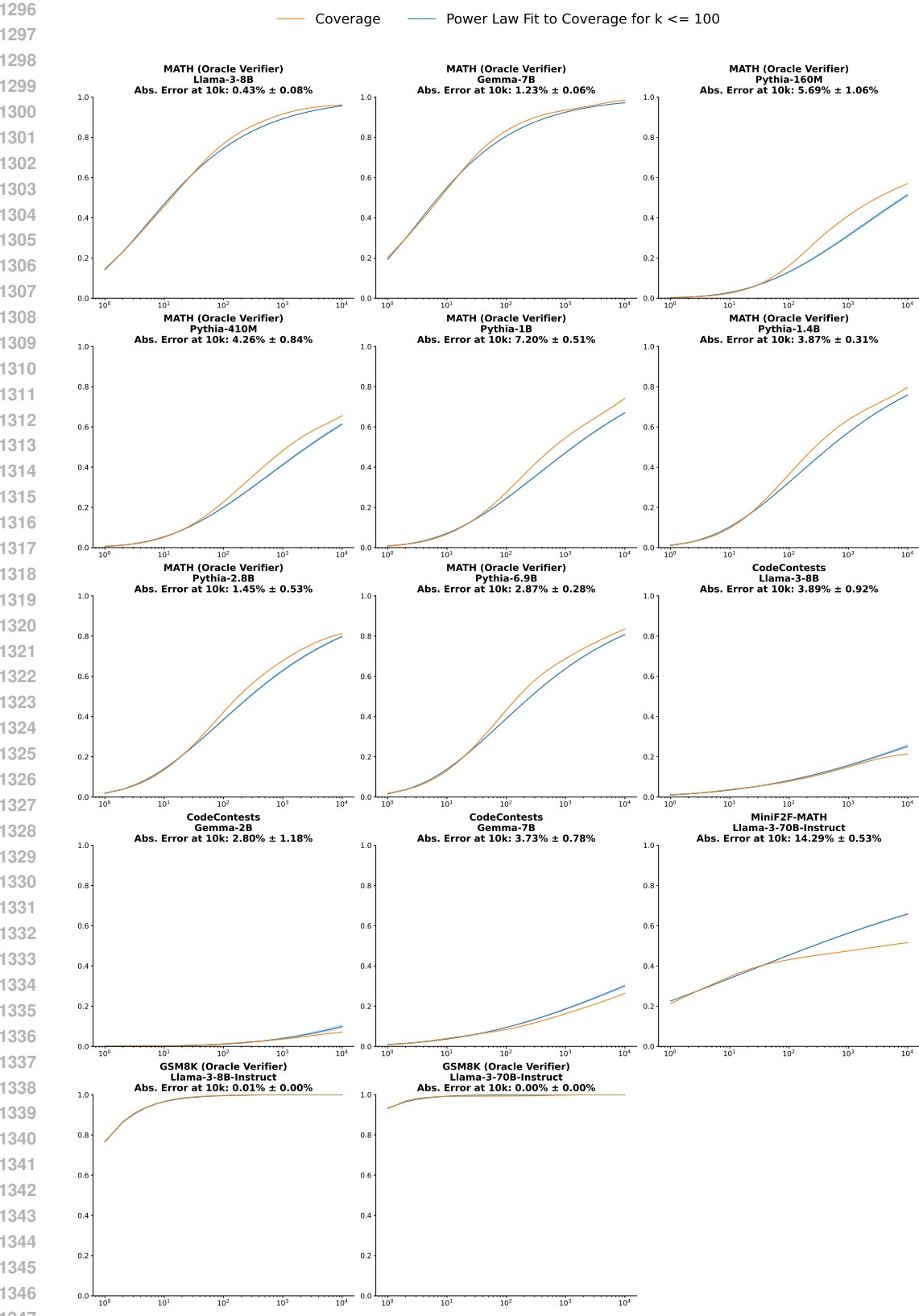

Figure 11: Predicting coverage values for high $k$ values by fitting a power law to the coverage curve for $k \leq 100$. Note that we recalculate the pass@k values used to do the power law fitting using a random $1k$ subset of the datapoints. We do this for 5 random subsets and report the mean and standard deviation of the pass@10k predictions for each subset.

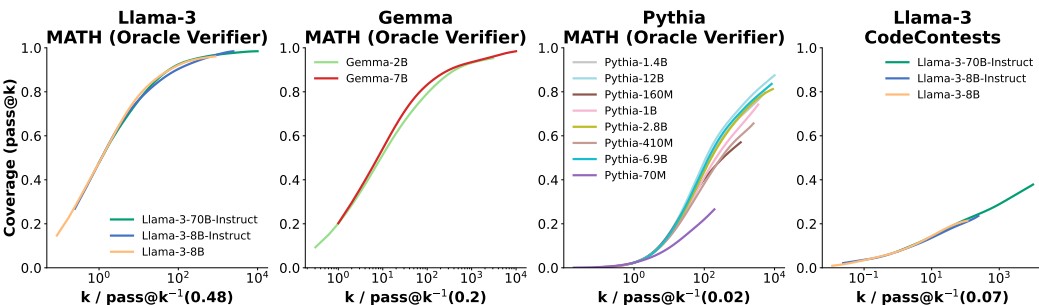

Figure 12: Overlaying the coverage curves from different models belonging to the same family. We perform this overlay by horizontally shifting every curve (with a logarithmic x-axis) so that all curves pass through the point $(1, c)$. We pick $c$ to be the maximum pass@1 score over all models in the plot. We note that the similarity of the curves post-shifting shows that, within a model family, sampling scaling curves follow a similar shape.

approach to verifying a piece of code and are not as comprehensive as methods like proof checkers. These imperfections in the verification process can lead to false positives and/or false negatives that are important to consider when applying repeated sampling. Below we provide two examples of software verifier imperfections that we encountered when generating our results from Section 2.1.

### F.0.1 FLAKY TESTS IN SWE-BENCH LITE

When producing our results on SWE-bench Lite, we identified that 11.3% of problems have flaky test suites that do not produce consistent results when running them on the same candidate solution. These flaky tests occasionally classify even the dataset's ground-truth issue solutions as incorrect. Additionally, the test suites for some issues can be non-determinstic depending on the candidate solution. For example, two SWE-bench Lite issues involve manipulating Python sets, which are naturally unordered. The gold solutions for these issues explicitly order the items in the set and pass the test suites reliably. However, some model-generated candidate solutions do not impose such an ordering, and therefore pass the tests on some "lucky" runs and not others. In Appendix B, we list all of the problem IDs where we identified flaky tests. We also report our SWE-bench Lite results from Figure 2 with the problematic issues removed, finding similar results to our evaluations on the whole dataset.

### F.0.2 FALSE NEGATIVES IN CODECONTESTS

Each problem from the CodeContests dataset comes with a set of input-output test cases used to asses the correctness of solutions. These test cases are more comprehensive than those from earlier coding benchmarks like APPS (Hendrycks et al., 2021a), cutting down on the frequency of false positive solutions that pass all test cases but do not fully solve the described problem. However, the construction of the CodeContests test suites leads to false negative solutions that are correct but fail the tests.

For some CodeContests problems, the problem description allows for multiple distinct correct outputs for a given test input. However, the corresponding test cases do not handle these scenarios, instead requiring that one particular correct output is emitted. Additionally, many CodeContests test cases have been programmatically generated by mutating original test cases from the problem. Some mutated inputs violate the problem's input specifications (e.g. a mutated input being zero when the description promises a positive integer). These malformed test cases can lead to inconsistent behaviour between different correct solutions.

We assess the prevalence of these issues by running each problem's test suite on the list of correct solutions that CodeContests provides. Of the 122 problems in the test set that have Python3 solutions, we find that 35 problems have "correct" solutions that fail the corresponding tests. Since we do not allow models to view all of a problem's test cases (and their peculiarities), applying repeated

sampling to these problems contains an element of "rolling the dice" to generate a solution that is not only correct, but emits the particular outputs that pass the tests.

## G    GSM8K INCORRECT ANSWER

As discussed in 4, we identify that a problem in the GSM8K test set (index 1042 on HuggingFace) has an incorrect ground truth solution.

> **Question**
>
> Johnny's dad brought him to watch some horse racing and his dad bet money. On the first race, he lost $5. On the second race, he won $1 more than twice the amount he previously lost. On the third race, he lost 1.5 times as much as he won in the second race. How much did he lose on average that day?

> **Answer**
>
> On the second race he won $11 because $1 + 5 \times 2 = << 1 + 5 * 2 = 11 >> 11$
> On the third race he lost $15 because $10 \times 1.5 = << 10 * 1.5 = 15 >> 15$
> He lost a total of $20 on the first and third races because $15 + 5 = << 15 + 5 = 20 >> 20$
> He lost $9 that day because $11 - 20 = << 11 - 20 = -9 >> -9$
> He lost an average of $3 per race because $9/3 = << 9/3 = 3 >> 3$
> #### 3

The mistake is in the second line of the answer: on the third race, Johnny's dad lost $16.5, not $15, meaning he made $11 and lost $16.5 + $5 = $21.5. So, the answer is an average loss of $3.5 per race, not $3 per race (the answer in the dataset).

## H    VERIFICATION USING A PROCESS REWARD MODEL

In Section 4, we benchmark three verification methods (majority voting, using a reward model with Best-of-N selection, and weighted majority voting using the reward model scores) on their ability to identify correct solutions from large sample collections. For the latter two methods in those experiments, we use ArmoRM-Llama3-8B-v0.1, which is an outcome reward model. In Figure 13, we extend those results to include an open source process reward model (PRM): math-shepherd-mistral-7b-prm Wang et al. (2024c). We follow Wan et al. (2024), and assign the score of the sample as the minimum score over all step-level rewards. We then use these reward model scores to select the final answer in two ways:

- PRM + Best-of-N: We choose the sample with the highest overall score.
- PRM + Majority Voting: We calculate a majority vote where each sample is weighted by its reward model score.

Due to resource constraints, we run this experiment on a 1k sample subset of the 10k samples generated by Llama3-8B-Instruct on the MATH dataset. We see that although coverage continues to increase up to 1k samples, the performance of the PRM with both methods plateaus before 100 samples similar to the previous verication methods.

## I    ADDITIONAL SAMPLING ABLATIONS

In Figure 14, we ablate the effect of the temperature and top-p values used for repeated sampling. In both sweeps, we sample 1k samples for the same 128 subset of the MATH dataset using Llama3-8B-Instruct. For the top-p sweep, we set temperature to $0.6$ and sweep over top-p values in $\{0.5, 0.75, 0.8, 0.9, 0.95, 1.0\}$. We see that the results are not very sensitive to temperature, with only top-p=0.5 being noticeably worse than the rest. For the temperature sweep, we set top-p to $1.0$ and sweep over temperature values in $\{0.4, 0.6, 0.8, 1.0, 1.2, 1.4\}$. We see that temperatures 1.2 and 1.4 have a significantly lower coverage than the rest.

## Llama-3-8B-Instruct - MATH

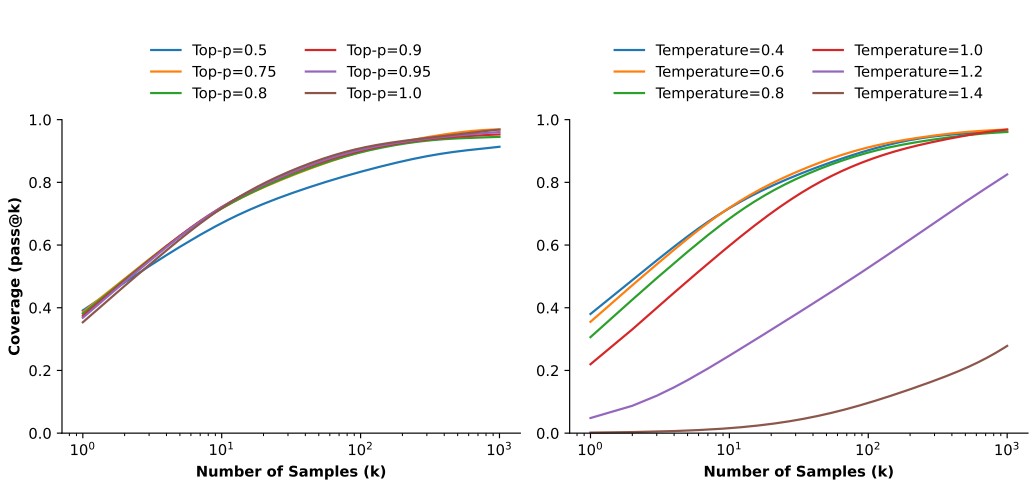

Figure 13: Comparing coverage (oracle selection) to the performance of selecting using a process reward model (PRM). Although coverage increases over all sample budgets, the performance with the PRM quickly plateaus before 100 samples.

Figure 14: Ablating the effect of top-p and temperature in repeated sampling. For the sweep over top-p values, we fix temperature to $0.6$ and for the temperature sweep we fix top-p to $1$.

