# OpenReview forum: "Large Language Monkeys: Scaling Inference Compute with Repeated Sampling"
_ICLR.cc/2025/Conference — Submitted to ICLR 2025_

### Official Review · Reviewer_NSZC · 2024-11-02

**Soundness:** 2
**Presentation:** 3
**Contribution:** 1
**Rating:** 3
**Confidence:** 4

**Summary:**

This paper studies the potential of scaling inference compute of LLMs. The authors show that, by generating repeated samples from language models, we can increase the “coverage”, i.e., the fraction of problems that are solved by any generated sample. The improved coverage directly translates to better performances when an oracle verifier is available.  The author also conduct experiments in the setting without oracle verifiers and find that the performances plateau quickly given repeated samples.

**Strengths:**

- The research problem is interesting and well-motivated. Scaling up training compute has led to remarkable success in deep learning. It is important to consider scaling inference compute.
- The paper is easy to read.
- The evaluation covers 5 different benchmark datasets and show consistent trends.

**Weaknesses:**

- One main claim, **scaling inference compute through repeated sampling leads to large improvements in coverage, seems trivial**. I believe that this fact is generally known in the community. Empirically, it has been observed in prior works like [1,2]. Mathematically, it is a simple consequence of equation 1. It is easy to prove that pass@$k$ monotonically increases with $k$ as long as there exists some $C_i>0$. The scaling curves are simply numerical calculation of equation 1 (correct me if this is not the case!). Thus, the novelty of this analysis seems limited.

- The paper **lacks in-depth analysis on scaling laws**. In Section 3, the proposed formula (equation 2) is simply adopted from the GPT-4 technical report. Then the “curve fitting” directly fit the power law to the curve generated by a _known formula_ (i.e., equation 1). I believe that meaningful scaling laws should be distilled from experimental observations. It’s not clear to me what is the insight of fitting some curves when the underlying formula is already known.

- I like the analysis on domains without automatic verifiers since it is a more realistic setting than the experiments based on oracle verifiers.  However, this section is very short and **lacks a deeper exploration of verifiers**. The conclusions here are based on experiments with a single existing 8B verifier. Strengthening this analysis with verifiers of varying sizes and other well-known verification approaches (e.g., process supervision [3]) would significantly enrich this section.

- Minor issues: Line 394: $k$ should be italic.

[1] Program Synthesis with Large Language Models

[2] Competition-level code generation with AlphaCode

[3] Let’s Verify Step by Step

**Questions:**

Please refer to the **weakness** part.

---

### Official Review · Reviewer_W6hC · 2024-11-02

**Soundness:** 2
**Presentation:** 4
**Contribution:** 2
**Rating:** 5
**Confidence:** 4

**Summary:**

This paper presents a systematic evaluation of repeated sampling for LLM generation across models and benchmark problems considering both oracle verifiers (pass@k) and reward models to select answers. The paper finds smooth scaling behavior with number of samples across all settings with oracle verifiers and proposes a parametric for the inference time scaling laws, even showing that repeated sampling from smaller models can outperform FLOP-matched sampling from larger models. Finally, the paper finds that there remains a large gap between the performance of oracle verifiers and reward models or other mechanisms like majority voting.

**Strengths:**

1. The paper provides a clear, rigorous study of an important and basic topic in LLM inference. The paper tests many different benchmarks and base models and finds that the main results to be robust. A lot of these may exist in prior work, but not in a coherent and systematic way as presented here.

2. The paper is well-written and easy to follow.

**Weaknesses:**

1. The paper needs to be careful about claims of novelty. The paper does discuss some related work in the intro, but could be more clear that existing work has made many of these observations. The main diffference is that this paper gives a more systematic analysis. For example alphacode [1] and others have figures that look almost identical scaling figures (Fig 6 in alphacode paper), and [2] has similar figures about scaling samples with reward models.

2. I think the scaling laws presentation is missing the somewhat basic discussion that this is totally the expected behavior for computing pass@k for independent bernoulli samples, no LLMs needed. For example, try running this numpy code and you will reproduce the same kinds of scaling law curves on a log scale:
```
p = 0.001
T = 10000
n_trials = 1000

samples = np.random.random((n_trials, T)) < p
passes = np.cumsum(samples, axis=1) >= 1
pass_rate = np.mean(passes, axis=0)
```
Of course this is a simplification since there should be a different value of p for each problem in the test set, but the basic idea is that when you take independent samples of a bernoulli variable, this is the expected behavior. There is likely a closed form for this simple process. It would probably make sense to model this directly rather than fitting a heuristic scaling law.

3. It is not clear how the presented scaling laws could be useful. The usual usefulness of pre-training scaling laws is to predict the optimal model size at a FLOP budget beyond those used for training. This kind of extrapolation is not tested in the paper. Moreover, the scaling laws are different for each specific model/task combinations and seem cheap to estimate, so it is not clear how prescriptive they are.

4. The stated conclusions about verifiers seem to be too strong given that the paper only uses one reward model that is not particularly tailored to the problems studied. The off-the-shelf RM is chosen for performance on reward bench reasoning, but to my understanding this benchmark is mostly about humaneval-style coding and then the RM is being applied on math reasoning problems. For example, [2] trains task-specific RMs and does not observe the same sort of saturation.

5. Some discussion of related work is missing when discussing how to improve verifiers in the future work directions. For example, [3] proposes an objective for reward modeling to allow LMs to evaluate themselves with a linear model on their representations and [4] uses models to evaluate themselves.

[1] Li, Yujia, et al. "Competition-level code generation with alphacode." Science 378.6624 (2022): 1092-1097.

[2] Lightman, Hunter, et al. "Let's verify step by step." arXiv preprint arXiv:2305.20050 (2023).

[3] Li, Kenneth, et al. "Q-Probe: A Lightweight Approach to Reward Maximization for Language Models." arXiv preprint arXiv:2402.14688 (2024).

[4] Yuan, Weizhe, et al. "Self-rewarding language models." arXiv preprint arXiv:2401.10020 (2024).

**Questions:**

See weaknesses

---

### Official Review · Reviewer_zJ2q · 2024-11-05

**Soundness:** 3
**Presentation:** 3
**Contribution:** 2
**Rating:** 6
**Confidence:** 4

**Summary:**

This work considers the question of scaling inference compute through repeated sampling.  The authors find that repeated sampling can greatly improve the performance on reasoning/code tasks, particularly when there is an external verifier that can be used to check the result.  At a high level, this work targets an interesting research question and the methodology looks sound.  However, parts of the paper are poorly written/confusing (particularly in explicating the experimental setup + results with/without the verifier), and could benefit from some major revisions.  I am willing to raise my score if the authors address my questions/concerns.

**Strengths:**

* While it isn't surprising that repeated sampling improves performance, the authors have quantified this phenomena in a precise way.  In particular, they find a relatively clean scaling relation between the coverage and number of samples.
* There is good diversity in the models and datasets used, from more "agentic" tasks like SWE-bench to standard math/code reasoning.

**Weaknesses:**

* The authors should include a few more details about the experimental setup or streamline the existing writing.  For example, one may have chosen a couple of different approaches on how to incorporate the verifier: the first is an iid setup, where the verifier is just used to pick the final answer out of several of attempts.  The second is where the model is provided some verifier signal (e.g., is the answer right/wrong?) between attempts, and asked to review its reasoning trace.  Both scenarios interesting, but there were some experimental choices made in the paper that bear some justification.
* The improved performance with repeat sampling is not meaningful in and of itself; if you have enough attempts, the fraction that you will get right will of course go up.  However, it may be useful for the reader to show an example of repeat sampling where the model gets the right answer after a small number of samples -- which part of the reasoning trace gets "fixed" between samples typically?  Can we characterize the mistakes that the models make better?
* I am a bit unclear on the relevant baseline here.  How should we compare spending the inference compute on sampling (versus, say, doing X-of-thought approaches)?
* A common assumption in the literature is that verification is easier than generation.  Interestingly, this work seems to show that while non-oracle verification can provide some wins, the improvement is overall quite small.  It would be interesting to dig into this question a bit more, and study some explicit examples where the reward model (or majority vote) fails.  How much of the lack of improvement is due to the inherent noisiness of the reward model?  Can the authors characterize the failure mode here a bit more precisely (is non-oracle verification almost as hard as generation in this case)?

**Questions:**

(See above)

---

### Official Review · Reviewer_feio · 2024-11-12

**Soundness:** 2
**Presentation:** 2
**Contribution:** 2
**Rating:** 6
**Confidence:** 5

**Summary:**

The paper studies scaling laws for a new axis of compute for LLMs: inference time compute. They empirically find that coverage (pass@N) improves with the number of inference time samples log-linearly and can be modeled with an exponentiated power law.
In domains like coding, where automatic verifiers exist, i.e., verification is much easier than generation, the performance improves from 15.9% with one sample to 56% with 250 samples, outperforming the single-sample state-of-the-art of 43%. In domains lacking automatic verifiers, the authors observe that typical approaches for selecting from a set of IID sampled responses, such as majority voting and reward models, reach a performance plateau after several hundred samples and do not effectively scale with an increasing sample budget, and thus the performance on these problems is bottlenecked by the accuracy of the verifier.

**Strengths:**

- The paper presents an extensive analysis of coverage on math and coding tasks and plots empirical scaling laws (and fitted ones) on a multiple model families like Gemma, Pythia and Llama. One of the most interesting trends to note is that models from the same family only affected the offset of the log-linear plot, and not the slope. This is different from typical scaling laws for pre-training loss seen in prior works.
- The authors also extend analysis to inference time compute, and show that sometimes drawing multiple samples from a smaller (and less capable model) is more compute-efficient (in terms of FLOPs), compared to a single sample from a larger and more capable model. This analysis can be immediately used to reduce the inference time for models deployed in practice, without any loss in performance, at least when a reliable verifier is available.
- The analysis in Figure 6,7 is particularly compelling to put more effort in improving verification, since it suggests that most test-time methods for reasoning/safety etc., are not failing due to lack of coverage, but more due to the inaccuracy of the verifier.

**Weaknesses:**

- While the coverage analysis expands on different models and tasks, it does not model the affect of other common parameters of interest like temperature, top-K etc.
- Some recent works like (Snell et al. 2024, Setlur et al. 2024) show that beam search against an automated verifier improves compute efficiency, with a fixed beam size. Having analysis on this direction of compute use would make the paper stronger.
- While the authors fit scaling laws for coverage, it would be much more useful to see if scaling laws can also be fit for the setting with trained verifiers, that may not be perfect, accounting for the error in the verifier. Currently it is unclear how the size/data used for the trained verifier affects the compute scaling laws for best-of-n inference.
- In general, the paper presents interesting results for best-of-N inference, but they are quite narrow and expected. Broadening the laws to other forms of compute usage, or identifying how the "learnability" of a verifier affects these laws can help to make the work more complete.

[1] Scaling LLM Test-Time Compute Optimally can be More Effective than Scaling Model Parameters (C. Snell, J. Lee, K. Xu, A. Kumar)
[2] Rewarding Progress: Scaling Automated Process Verifiers for LLM Reasoning (A. Setlur, C. Nagpal, A. Fisch, X. Geng, J. Eisenstein, R. Agarwal, A. Agarwal, J. Berant, A. Kumar)

**Questions:**

- Maybe I missed discussion on this but is there any attempt to also parameterize the coverage scaling law in terms of the model size?
- Can the authors compare their work with parallel work from Snell et. al. 2024 (Scaling LLM Test-Time Compute Optimally can be More Effective than Scaling Model Parameters)? In particular it would be nice to see discussion on whether the exponentiated power law is also a reasonable to fit scaling curves generated by forms of test-time compute methods, like sequential corrections explored by Snell et al.

---

### Meta-Review · Area_Chair_CBd7 · 2024-12-23

**Metareview:**

This paper studies scaling laws for LLMs as a function of the inference time budget. It makes empirical observations that coverage improves with increasing inference time budget and can be modeled with an exponentiated power law. For domains with automatic verifiers (e.g., code generation), the paper shows significant improvement in performance with increased inference time budget. For domains without automatic verifiers, the paper shows that performance plateaus after few hundred samples due to inaccurate verifier.

The reviewers liked the experimental findings acknowledging that most of the findings are known to the community. The only non-trivial contribution is the scaling curve and its claimed predictive power. However, two reviewers noted that this part is not well done and the analysis is somewhat insufficient. I agree with this observation.

Therefore, I'm recommending to reject this paper and strongly encourage the authors' to improve the scaling curve and its analysis based on the feedback from reviewers' for resubmission.

**Additional Comments On Reviewer Discussion:**

Summarized in meta review.

---

### Decision · Program_Chairs · 2025-01-22

Reject